# Prevalence and Abundance of Bacterial Pathogens of Concern in Shrimp, Catfish and Tilapia Obtained at Retail Stores in Maryland, USA

**DOI:** 10.3390/pathogens12020187

**Published:** 2023-01-25

**Authors:** Salah Elbashir, Michael Jahncke, Angelo DePaola, John Bowers, Jurgen Schwarz, Anuradha J. Punchihewage-Don, Byungrok Min, Tom Rippen, Salina Parveen

**Affiliations:** 1Food Microbiology and Safety Laboratory, Department of Agriculture, Food and Resource Sciences, University of Maryland Eastern Shore, Princess Anne, MD 21853, USA; 2Virginia Seafood Agricultural Research and Extension Center, Virginia Tech., Hampton, VA 23669, USA; 3Angelo DePaola Consulting, 12719 Dauphin Island Pkwy, Coden, AL 36523, USA; 4U.S. Food and Drug Administration, College Park, MD 20740, USA

**Keywords:** prevalence, *Vibrio*, *Salmonella*, *Campylobacter*, *E. coli*, shrimp, tilapia, catfish

## Abstract

Outbreaks of human gastroenteritis have been linked to the consumption of contaminated domestic and imported seafood. This study investigated the microbiological quality of seafood obtained from retail stores on the Eastern Shore of Maryland. A total of 440 samples of domestic and imported frozen shrimp, catfish and tilapia samples were analyzed for aerobic plate count (APC), total coliforms, *Escherichia coli* and seafood-borne-pathogens (*Vibrio parahaemolyticus, Vibrio vulnificus, Salmonella*, *Campylobacter jejuni*). The prevalence of APC, coliforms and *E. coli* positive samples was 100%, 43% and 9.3%, respectively. Approximately 3.2%, 1.4%, 28.9% and 3.6% of the samples were positive for *V. parahaemolyticus, V. vulnificus, Salmonella* and *Campylobacter jejuni*, respectively. The MPN/g ranges were 150–1100 MPN/g for vibrios, 10–1100 MPN/g for *Salmonella* and 93–460 MPN/g for *C. jejuni* in seafood, respectively. Comparing bacterial prevalence by type or source of seafood, the only significant difference identified was *Salmonella*-positive imported tilapia (33.3%) versus domestic tilapia (19.4%). The quantitative data on pathogen levels in the present study provide additional information for quantitative risk assessment not available in previous surveys. The findings of this study suggest the association of potential food safety hazards with domestic and imported seafood and warrant further large-scale studies and risk assessment.

## 1. Introduction

Global seafood production has been increasing dramatically during the last three decades [1,2]. Approximately 88% of this production was directly consumed by humans [1]. China is the leading seafood producer (38.5% of global production) and is the foremost consumer worldwide [3]. The U.S. is rated as the second largest consumer of seafood based on the United Nations Food and Agriculture Organization (FAO) consumption statistics from 2011 to 2013 [3]. Shrimp, canned tuna, salmon, tilapia, Alaska pollock, pangasius, crab, cod, catfish and clams were the top ten and most favored seafood products consumed in the U.S. in 2016 [4]. Ninety seven percent of the fish and shellfish are imported from overseas [5,6]. Nearly 50% of U.S. seafood imports were produced by aquaculture. Frozen seafood accounts for approximately 75% of the gross imports. Increased consumption and importation of seafood raise concerns over its safety [4,7].

The U.S. Centers for Disease Control and Prevention (CDC) reported that 53% of the seafood multistate outbreaks in 2014–2015 were caused by bacteria, 37% from viruses, 0.9% from parasites and 7% from chemical contaminants [8,9]. Iwamoto et al. [10] reported that bacteria caused 76% of the seafood outbreaks, 21% by viruses and approximately 3% by parasites. Lipp and Rose [11] reported that a third of foodborne outbreaks had seafood as the main etiologic vehicle. 

Pathogenic *Vibrio* spp., *Vibrio cholerae, Vibrio parahaemolyticus* and *Vibrio vulnificus* cause outbreaks and sporadic illnesses usually associated with the consumption of raw or undercooked seafood [10,12,13]. *Salmonella* is the second most common causative bacterial agent of illnesses in the U.S. food supply and frequently implicated in multistate outbreaks [8,9,10,13,14,15,16]. Seafood-associated illnesses have led authorities to implement a zero-tolerance level for *Salmonella*. *Campylobacter jejuni* and *Campylobacter coli* are the leading cause of bacterial gastrointestinal infections in humans. Fish and shellfish are incriminated as an etiological food that may transmit *C. jejuni* to humans, which causes gastroenteritis [17]. Recently reported seafood-associated outbreaks have heightened concerns for these pathogens [18]. Currently, other than results of inspections reported by USFDA, adequate data on the prevalence of foodborne pathogens in domestic and imported seafood are not available. The objective of this study was to survey the prevalence and abundance of bacterial indicators, *V. parahaemolyticus*, *V. vulnificus*, *Salmonella and Campylobacter,* in imported and domestic seafood obtained at retail stores on the Eastern Shore of Maryland.

## 2. Materials and Methods

### 2.1. Collection of Samples

A total of 440 frozen domestic and imported seafood samples (shrimp (whole), catfish (fillet) and tilapia (fillet)) were aseptically and randomly collected at monthly intervals from four retail stores on the Delmarva Peninsula on the Eastern Shore of Maryland from August 2012 to July 2013. The stores from which samples were collected were labeled as A, B, C and D, respectively. On each sampling day, up to 13 samples of each seafood type were collected from the four stores, depending on the availability. The type of seafood, country of origin and production date were also recorded during collection. The random collection of samples did not consider the duration and conditions of storage as well as the handling and transportation conditions due to the privacy of each store. All imported tilapia (n = 84) were from China while all imported catfish (n = 60) were from Vietnam. Imported shrimp originated from India (n = 20), Indonesia (n = 33) and Thailand (n = 29). 

Samples collected frozen included shrimp (n = 142), catfish (n = 142) and tilapia (n = 156). Each sample weighed 500 g. After collection, all samples were placed in an ice chest with ice packs and a data logger (ACR Systems, Inc., Surrey, Canada) and transported directly to the Food Microbiology and Safety Laboratory at the University of Maryland Eastern Shore for analysis. All samples remained frozen during transport and all analyses were performed within four hours of collection.

### 2.2. Preparation of Samples 

Upon arrival at the laboratory, samples were prepared according to Andrew et al. [19]. All samples were defrosted and aseptically cut into smaller pieces, placed in a plastic bag and hand massaged for homogenization. A homogenate was made from 500 g of sample, 200 g were saved for other experiments and, from the remaining portion, 25 g of each sample of catfish and tilapia and 50 g of shrimp were weighed and placed in a plastic bag containing 225 mL for catfish and tilapia and 450 mL for shrimp of buffered peptone water (Becton Dickinson Diagnostic Systems, Sparks, MD, USA) as per Andrews et al. [19]. The contents were massaged and stomached for 5 min and 20 mL of the rinse fluid was transferred to individual 50 mL screw-cap tubes including a portion of the sample (<10%) for subsequent microbiological analysis. The remaining samples were stored at −20 °C for quantitative experiments. 

### 2.3. Aerobic Plate Count—APC and Coliforms/Escherichia coli Counts

Samples were analyzed for aerobic plate count (APC), total coliforms and *E. coli*. The 3M total colony count Petri-films were used according to manufacturer instructions (Carolina Biological Supply Company, New York, NY, USA). One milliliter of the sample was inoculated on each Petri-film and incubated at 35 °C for 24 h. After incubation, colonies that appeared as pinkish red on Petri-films were counted and enumerated as CFU/g. The APC is indicative of the general quality as well as of the quality of handling and storage procedures [10].

The 3M total coliforms/*E. coli* Petri-films were used according to manufacturer instructions (Carolina Biological Supply Company, New York, NY, USA). One milliliter of the sample was inoculated on each Petri-film and incubated at 37 °C for 24 h. Red colonies with or without gas production and blue to red-blue colonies associated with gas bubbles were considered as coliforms and *E. coli*, respectively. These colonies were counted and expressed as log CFU/g of sample. The existence of coliforms and/or *E. coli* in marine food products is evidence of contamination from a terrigenous source [10]. 

### 2.4. Total Vibrio, V. parahaemolyticus and V. vulnificus

Total *Vibrio* and *Vibrio* species were isolated using a method described by Kaysner et al. [20]. Briefly, 25 g of the catfish or tilapia samples and 50 g of shrimp samples were added to 225 and 450 mL of alkaline peptone water (Becton Dickinson Diagnostic Systems, Sparks, MD, USA), respectively, stomached for 2 min and incubated at 35 °C for 24 h. The alkaline peptone water cultures were streaked onto thiosulfate citrate bile salts sucrose (TCBS) agar and modified cellobiose-polymyxin β-colistin agar (mCPC) (Becton Dickinson Diagnostic Systems, Sparks, MD, USA) plates for the isolation of *V. parahaemolyticus* and *V. vulnificus*, respectively. The plates were incubated at 35 °C for 24 h for TCBS agar and 40 °C for 24 h for mCPC agar. Five presumptive *Vibrio* colonies (green or bluish-green, round and 2 to 4 mm in diameter for *V. parahaemolyticus* on TCBS and flat yellow, fried-egg shape and 1 to 2 mm in diameter for *V. vulnificus* on mCPC) were picked for confirmation using BAX-real time PCR assay according to the manufacturer’s instructions (Qualicon Diagnostic, Camarillo, CA, USA). Total *Vibrio* levels were calculated by adding green and yellow colonies on TCBS plates. 

### 2.5. Salmonella

*Salmonella* was isolated following a previous method [19] with some modifications. In brief, aseptically, 25 g of each catfish and tilapia sample and 50 g of each shrimp sample were added to 225 mL and 450 mL of sterile lactose broth (Becton Dickinson Diagnostic Systems, Sparks, MD, USA), respectively, stomached for 2 min and incubated for 24 ± 2 h at 35 °C. For the second enrichment, 10 mL of Rappaport-Vassiliadis broth (RV) (Becton Dickinson Diagnostic Systems, Sparks, MD, USA) was inoculated with 1 mL of the previously incubated lactose broth mixture and incubated at 35 °C for 24 h. Enriched samples were streaked onto Xylose Lysine Trigitol 4 (XLT4) (Becton Dickinson Diagnostic Systems, Sparks, MD, USA) agar plates. Five colonies with typical *Salmonella* phenotypes (black colonies) were confirmed by BAX-PCR assay for *Salmonella.*

### 2.6. Campylobacter jejuni

According to Hunt et al. [21], aseptically, 25 g of catfish and tilapia and 50 g of shrimp were added to 225 mL and 450 mL of the Bolton enrichment broth (Malthus Diagnostics, North Ridgeville, OH, USA) with lysed horse blood and antibiotic supplement (Malthus Diagnostics, North Ridgeville, OH, USA), respectively. The tightened containers were incubated at 37 °C for 4 h and then at 42 °C for 24 h. All subsequent incubations were conducted at 37 °C under microaerophilic conditions (85% N_2_, 10% CO_2_ and 5% O_2_). The Bolton broth cultures were inoculated onto modified Campy blood-free agar (mCCDA) (Malthus Diagnostics, North Ridgeville, OH, USA) by using a cotton-tipped swab followed by streaking. After 48 h of incubation, five presumptive *Campylobacter* colonies (small, grey, moist and flat spreading or drop-like) per sample were confirmed using *C. jejuni/C. coli/C. lari* BAX-real time assay according to the manufacturer’s instructions. 

### 2.7. Confirmation of Presumptive V. parahaemolyticus, V. vulnificus, Salmonella and Campylobacter Isolates Using BAX PCR

*V. parahaemolyticus* and *V. vulnificus* colonies were confirmed using *Vibrio cholerae/parahaemolyticus/vulnificus* BAX-real time PCR assay according to the manufacturer’s instructions (Qualicon Diagnostic, Camarillo, CA, USA). In brief, 10 mL of alkaline peptone water (APW) (Becton Dickinson Diagnostic Systems, Sparks, MD, USA) were inoculated with the presumptive *V. parahaemolyticus* and *V. vulnificus* isolates and incubated at 37 °C and 40 °C, respectively, for 24 h each. Five microliters of each incubated isolate were added to a 200 µL lysis reagent cluster tube (lysis reagents were prepared by adding 150 µL of protease to 12 mL lysis buffer). Tubes were heated at 37 °C for 20 min and then at 95 °C for 10 min. Tubes were cooled at 2 °C–8 °C for 5 min. Thirty microliters were transferred into each PCR tube and placed in a chilled cooling block. The tubes were placed in the cycler-detector and the samples were automatically run according to the manufacturer’s protocol (Qualicon). Within 1–2 h, the PCR amplified a DNA fragment that is specific to the target. After amplification, the automated system begins a detection phase in which the fluorescent signal is measured. The results were displayed as positive or negative symbols. A positive *V. parahaemolyticus* or *V. vulnificus* isolate was indicated by a sigmoid curve with a cycle threshold (Ct) from 20 to 40. 

For *Salmonella* isolates’ confirmation, 10 mL of lactose broth tubes were inoculated with the presumptive *Salmonella* and incubated at 37 °C for 24–48 h. Lysis, heating, cooling and detection of incubated isolates were performed as described for *V. parahaemolyticus* and *V. vulnificus*. A positive *Salmonella* isolate was indicated by a sigmoid curve with a cycle threshold (Ct) from 20–40. 

For confirmation of *C. jejuni*, in an air-tight tube, 10 mL of double strength Bolton broth with supplement (no blood) were inoculated with the presumptive *C. jejuni* and incubated micro-aerobically at 42 °C for 24–48 h. Lysis, heating, cooling and detection of incubated isolates were performed as described for *V. parahaemolyticus, V. vulnificus and Salmonella.* A positive *C. jejuni* isolate was indicated by a sigmoid curve with a cycle threshold (Ct) from 20 to 40.

The lower limit of detection for qualitative methods are 1 cell/25 g for all pathogens in tilapia and catfish and 1 cell /50 g of all pathogens in shrimp. Confirmed for *V. parahaemolyticus* and *V. vulnificus* isolates were stored at −80 °C in trypticase soy broth (TSB) and 1% NaCl with 24% glycerol (Becton Dickinson Diagnostic Systems, Sparks, MD, USA); confirmed *Salmonella* and *Campylobacter* isolates were stored at −80 °C in brain heart infusion broth with 24% glycerol (Becton Dickinson Diagnostic Systems, Sparks, MD, USA) for further analysis.

### 2.8. Most Probable Number (MPN)

Samples that were positive for bacterial pathogens were enumerated using the three tube Most Probable Number (MPN) method as described by Blodgett, [22]. All samples were analyzed for MPN within 48 h. of collection as the BAX PCR system allowed detection and confirmation of isolates within 20 h. In brief, 10-fold serial dilutions (0.1–0.00001) of each sample were made in Phosphate Buffer Saline (PBS) (Becton Dickinson Diagnostic Systems, Sparks, MD, USA) and 1mL of each dilution was transferred to 3 × 10 mL of alkaline peptone water (APW), 3 × 10 mL of buffered peptone water (BPW) and 3 × 10 mL of Bolton broth for *V. parahaemolyticus* and *V. vulnificus*, *Salmonella*, *C. jejuni*, respectively, and incubated at 37 °C for 18–24 h. Secondary enrichment for MPN was only conducted for the enumeration of *Salmonella*. Bacterial growth was determined by turbidity, then turbid tubes were streaked onto selective media for each pathogen. After incubation at 37 °C for 24 h, presumptive colonies were picked and confirmed using BAX-PCR as described in previous sections. Based on the presence/absence of confirmed colonies on each plate, MPN/g of sample was calculated using standard U.S. Food and Drug Administration (FDA) procedures and table (https://www.fda.gov/food/foodscienceresearch/laboratorymethods/ucm109656.htm, accessed on 23 January 2022). 

### 2.9. Serotyping of Salmonella

*Salmonella* serovars were determined by the USDA’s National Veterinary Services Laboratories (Ames, IA, USA). The *Salmonella* somatic O antigen and the flagellar H antigen combination codes were tested against different antisera to subtype the organism [23]. The *Salmonella* somatic O antigen has been determined with corresponding antisera in the Bacterial Identification (BI) section of the Diagnostic Bacteriology and Pathobiology Laboratory (DBPL) at the National Veterinary Services Laboratories (NVSL) [24]. The flagellar H antigen was determined after the determination of the surface O antigen [25].

### 2.10. Data Analysis

Measurement outcomes were evaluated using one-way ANOVAs, *t*-tests or Pearson correlation when quantitative, by the Fisher’s exact test when qualitative and by logistic regression when outcomes were both quantitative and qualitative. Aerobic plate counts were converted into log values (log CFU/mL) and sorted by seafood type and country of origin. Based on the APC limits established by the International Commission on Microbiological Specifications for Foods [21,26,27], the percentages of seafood samples that fell into the microbiological quality categories of good (<5 × 10^5^ CFU/g), marginally acceptable (5 × 10^5^ to 1 × 10^7^ CFU/g) and unacceptable (> 1 × 10^7^ CFU/g) quality were calculated. Log transformed APC data sorted by seafood type, fresh/frozen status sampling store and sampling date were analyzed by using analysis of variance (SAS for Windows, version 9; SAS Institute NC., Cary, NC, USA). Logistic regression was used to evaluate associations between level of total coliforms (log transformed) and the detection of *Vibrio* species, *Salmonella and C. jejuni* and, for samples with detected and enumerated levels of one or more pathogens, associations between the level of total coliforms (log transformed) and the log MPN/g for each pathogen were evaluated by Pearson correlation. Due to low prevalence of detectable *E. coli*, no correlation analysis was conducted between *E. coli* and pathogens. Microbiological qualities data and differences in prevalence for each bacterium were analyzed by Fisher’s exact test. An alpha level of 0.05 was used when determining statistical significance.

## 3. Results 

### 3.1. Aerobic Plate Counts (APC) and Microbiological Quality

All samples were positive for aerobic plate count (APC). The average log CFU/g of APC for each seafood type and source ranged from 1.8–2.0, 3.8–4.6 and 2.4–2.8 log CFU/g in shrimp, catfish and tilapia, respectively. Comparing domestic to imported, there were no statistically significant differences (*p* > 0.05) between average log CFU/g for APC (Figure 1). 

Based on the acceptable limits established by the International Commission on Microbiological Specifications for Foods (ICMSF) [27], four domestic shrimp and four domestic catfish samples were microbiologically unacceptable. Two domestic and six imported shrimp samples and nine domestic and seven imported catfish samples were reported as marginally acceptable. In addition, four domestic and seven imported tilapia samples were considered marginally acceptable (Table 1). Across the seafood types, the average APC values for both domestic and imported catfish (DCF and ICF) were significantly higher than for domestic and imported shrimp (DSH and ISH) and tilapia (DTA and ITA) by 2 logs or more (*p* < 0.05) (Figure 1).

### 3.2. Total Coliforms and E. coli 

Forty-three percent of all seafood samples were positive for coliforms (Figure 2) and 9.3% of them were positive for *E. coli*. With respect to seafood sources, prevalence of samples positive for total coliforms was significantly higher among imported shrimp (63.3%) compared to domestic shrimp (8.2%) (*p* < 0.05). In contrast, the differences between the prevalence of total coliform positive samples among domestic catfish (48%) and domestic tilapia (44%) compared to imported catfish (31%) and imported tilapia (47%) were not statistically significant.

The prevalence of samples positive for *E. coli* in each of the three seafood types were 3.6%, 14.0% and 2.8% in domestic shrimp, catfish and tilapia, respectively, compared to 8.2%, 13.3% and 10.7% in respective imported samples (Figure 3). Comparing imported to domestic, there were no statistically significant differences (*p* > 0.05) between the prevalence of samples positive for *E. coli* in shrimp and catfish while there was a significant difference in the prevalence of samples positive for *E. coli* in domestic and imported tilapia (*p* < 0.05). Approximately, 10% of the coliforms and *E. coli* positive samples were considered unacceptable for human consumption based on ICMSF standards (exceeding MPN of 330/100 g in a single sample or 230/100 g for two or more samples) [27]. 

### 3.3. Major Bacterial Pathogens of Concern

Exploratory data analysis indicated no significant seasonal pattern in the prevalence of pathogens. The overall prevalence of detectable total *Vibrio*, *V. parahaemolyticus and V. vulnificus* for all samples (n = 440) was 4.5%, 3.2% and 1.4%, respectively. All vibrios were isolated from shrimp samples. All domestic and imported catfish (n = 142) and tilapia (n = 142) samples were negative for total *Vibrio*. Fourteen percent of shrimp samples were positive for total *Vibrio*. For all shrimp (n = 71 domestic and 85 imported), a total of 14 (9%) and six (3.8%) samples were confirmed positive for *V. parahaemolyticus* and *V. vulnificus,* respectively. Four (5.6%) positive *V. Parahaemolyticus* were isolated from domestic shrimp and 10 (11.8%) from the imported respectively. In addition, three *V. vulnificus* isolates were isolated from domestic shrimp, another three from imported shrimp and prevalence of samples positive for total *Vibrio* for each type/source were 9.8% and 15.3%, respectively, as shown in Figure 4. Among samples confirmed positive (n = 20), the MPN/g ranged from 75 to 1100 for both *Vibrio* species and *V. parahaemolyticus* and 210 to 1100 for *V. vulnificus*.

A total of 127 (28.9%) samples from three types of seafood (n = 440) were confirmed positive for *Salmonella*. By seafood type, *Salmonella* was isolated as shown in Figure 5. Differences in the prevalence of *Salmonella* positives between domestic and imported samples were found in all types; shrimp (36.8% of domestic and 23.5% of imported), catfish (36.6% of domestic and 23.3% of imported) and tilapia (19.4% of domestic and 33.3% of imported). All *Salmonella* isolates (n = 127) were serotyped as *S.* Typhimurium var 5. For samples where *Salmonella* isolates were recovered, the MPN/g ranged from 10 to 1100 for both domestic seafood (n = 62) and imported seafood (n = 65). 

Sixteen of the 440 samples (3.6%) were confirmed positive for *C. jejuni*. Of these, nine were domestic samples including four shrimp (5.6%), two catfish (2.4%) and three tilapia (5.2%). Seven samples were imported including one shrimp (1.2%), five catfish (8.3%) and one tilapia (1.2%) (Figure 6). With respect to these major pathogens, no significant differences were observed between domestic and imported seafood except in catfish (2.4% of domestic and 8.3% of imported). For samples where *C. jejuni* isolates were recovered, the MPN/g ranged from 93 to 460 for domestic seafood and 210 to 460 for imported seafood.

In this study, we observed a statistical association between the level of total coliforms (log CFU/g) and the prevalence of detectable *Salmonella* (*p* < 0.001) but no association was evident between the level of total coliforms and the prevalence of either *Vibrio* or *Campylobacter*, which were both detected less frequently than *Salmonella*. On average, total coliforms levels were 0.1 logs higher when *Salmonella* was detected compared to when it was not. For the subset of samples with detected and enumerated pathogen level, no correlation was evident between the level of total coliforms (log CFU/g) and the log MPN/g of *Salmonella, Vibrio* or *Campylobacter*, which varied over a relatively narrow range. 

## 4. Discussion

Concerns about seafood microbiological quality and safety in the U.S. have increased due to reported illnesses and outbreaks. To our knowledge, this is the most comprehensive study demonstrating both the prevalence and abundance of major bacterial pathogens and indicators in imported and domestic seafood. Shrimp, catfish and tilapia were chosen for this study because of their popularity and preference among consumers in the USA [4].

The absence of statistically significant differences in average log CFU/g for APC between domestic and imported seafood suggests equivalent microbial quality and hygienic practices. While the muscle tissue of catfish and tilapia are normally sterile, bacteria that are abundant in the intestines, skin and gills can cross-contaminate fillets during processing. On the other hand, the intestines of unpeeled shrimp remain with the product. APC levels in the current study are considerably lower than reported in most previous studies. APC values previously reported for shrimp ranged from 4.8–7.1 log CFU/g and estimated ranges for microbiologically unacceptable samples were ≤10% [28,29,30,31] and 13.6% for finfish (catfish, salmon, tilapia and trout) in the USA [31,32]. Low APC values observed in the current study relative to previous surveys are likely to reflect hygienic processing practices and/or little time between processing and freezing, which prevents further bacterial growth.

The prevalence of total coliforms (43%) and *E. coli* (9.3%) in this study reflect the sanitation of production environments through the supply chain to retail. Similar prevalence of *E. coli* (9.4%) was reported in a survey of imported shrimp [32]. However, prevalence has varied significantly in other studies which reported prevalence below 7% and enumerated levels below 10 MPN/g [28,31]. *E. coli* was reported in higher prevalence in India and Vietnam because of poor seafood hygiene [7]. However, the prevalence rate of *E. coli* recovered from raw oysters was only 2.2%, and from dried squid products 14.3% [33]. Wang et al. [32] and other studies failed to detect *E. coli* in imported seafood. The higher prevalence of coliforms and the higher amounts of unacceptable samples in this study indicated poor hygienic conditions. The finding of Atwill and Jeamsripong [34] showed higher prevalence of *E. coli* among seafood (85%) with a total average concentration of 2 × 10^4^ (± 4 × 10^4^ MPN/g). In addition, the same study [34] determined the prevalence of fecal coliforms in seafood equal to 100% with total average of 9 × 10^4^ MPN/g (± 4 × 10^4^ MPN/g). Comparing domestic to imported seafood types in the current study, there were no statistically significant differences between them. The high prevalence of *E. coli* (14%) among samples of catfish from domestic and imported sources in this study suggested poor management, a heavy infestation of reptiles and amphibians and/or sewage contamination of the aquaculture [14]. On the other hand, low prevalence possibly indicated an improvement in the management and hygiene practices. Previous studies reported that 4.4–13.6% of shrimp and finfish samples were unacceptable in their microbiological quality [22,28,30,31,32]. With respect to domestic vs. imported seafood, microbiologically unacceptable samples from domestic shrimp and domestic catfish were within the range and matched the previous findings (>1 × 10^7^ CFU/g). However, none of the imported shrimp and catfish were unacceptable. Some of the domestic shrimp (<2%) and domestic catfish (9%) were determined as microbiologically marginally acceptable (Table 1). The percentages of marginally acceptable imported shrimp (7%) and catfish (12%) were within the same range as the previous studies above. However, 6% of the domestic and 8% of imported tilapia were microbiologically marginally acceptable. The findings in this study showed that high percentages of seafood samples of all types met the standard of microbiologically good quality. There were no statistically significant differences (*p* > 0.05) between the domestic and imported seafood of any type, which is consistent with the results of previous studies by Wang et al. [32] and Berry et al. [30] in imported seafood in the USA.

The prevalence was 4.5% for total *Vibrio* species, 28.9% for *S.* Typhimurium and 3.6% for *C. jejuni*. There was a large difference between imported and domestic *Salmonella* prevalence in shrimp, catfish and tilapia (36.8%, 36.6% and 19.4% of domestic and 23.5%, 23.3% and 33.3% of imported shrimp, catfish and tilapia, respectively). Some studies indicated that the prevalence of *Vibrio* species ranged between 0% and 63% [22,28,29,30,31,32]. The prevalence of *Salmonella* in a previous FDA study (1990–1998) was 10% [26]. The prevalence of *S.* Typhimurium was rated as 10th among different *Salmonella* serovars [26,35]. Previous studies in the USA revealed no *Salmonella* contamination in seafood [28,31]. In Japan, Hara-Kudo et al. [33] reported a prevalence of 0.5% *Salmonella* in raw oysters. However, some other studies reported prevalence rates of *Salmonella* ranging between 4.3–18% [7,29,32,35]. Prevalence of *C*. *jejuni* of less than 1% was reported in seafood in previous studies [10,32]. Different methods in these studies, all of which may not have the same limit of detection (LOD), are maybe the cause of this wide range. In comparison to other studies, the findings of this study varied among pathogens. Prevalence of *Vibrio* spp. (4.5%) in shrimps were among the lower rates compared to other studies which ranged from 0% to 63% in the USA [28,31]. The absence of fresh seafood and limiting the sample collection in this study to frozen seafood only may significantly impact the outcomes. 

*Salmonella* prevalence (28.9%) was significantly higher compared to previous studies, where *Salmonella* prevalence did not exceed 10% [26,32]. *C. jejuni* prevalence (3.6%) in this study was slightly elevated compared to other studies which reported around 1% prevalence recovered from seafood [32,36]. High rates of detection of *Salmonella* and *C. jejuni* in seafood is considered an alert for fecal contamination of aquaculture environments and/or cross-contamination [4]. Such facts explained the high prevalence of these two pathogens. The findings of this study reported *S.* Typhimurium as the only serovar recovered from all seafood regardless of the type and source. However, all isolates were genetically diverse when using pulsed-field gel electrophoresis (data not shown). There is no single explanation for this; however, it signals the need for future studies to determine a reasonable explanation. Other studies reported different serotypes isolated from raw oysters, such as Infantis, Schwarzengrund and Manhattan, as well as Typhimurium [33]. *S.* Typhimurium was isolated by Wang et al. [32] from imported tilapia. 

*Salmonella* prevalence in this study was much greater than reported by previous studies [26,29]. Sources of *Salmonella* in these seafood samples require additional studies to determine the role of pollution and natural aquatic occurrence, which is more difficult to control. With respect to type and source of seafood, the most notable difference between them was the prevalence of detectable *Salmonella* in domestic tilapia. 

In this study, the levels of *Vibrio*, *Salmonella and Campylobacter* were quantified whereas most previous studies only determined the prevalence of *Salmonella* and *Campylobacter* in seafood. Quantitative data improves the accuracy of risk assessment compared to previous data. The MPN ranged from 75 to 1100/g. *Vibrio* levels in the current study were much lower than reported in live oysters and blue crabs during the warmer months when MPN ranges from 10 to >100,000 [2,12,37]. This might be due to the collection and analysis of frozen samples instead of fresh/live samples. However, we were not able to compare the quantitative results of our study with previous studies on shrimp, tilapia and catfish due to lack of pathogen-specific quantitative data. Moreover, we did not find enough information on microbiological acceptability based on the level of specific pathogen *(Vibrio* spp., *Salmonella* and/or *Campylobacter*), though there is a plethora of information on seafood microbiological acceptability based on APC. We observed that none of the marginally acceptable and unacceptable samples were *V. parahaemolyticus, V. vulnificus* and/or *C. jejuni* positive. In contrast, only three samples from the marginally acceptable and none of the unacceptable were *Salmonella* positive. Based on APC, all *Salmonella* positive samples were acceptable for human consumption. However, based on FDA’s 2021 guidance for fish and fishery products, any sample with detectable *Salmonella* exceeds the established safety level for that organism and may be considered unacceptable for human consumption [37]. The findings of this study suggest that seafood should not be considered as safe for human consumption based only on APC levels.

Total coliform levels were weakly correlated with the prevalence of *Salmonella* in the seafood samples collected in this study. Total coliforms and *Salmonella* contamination can occur during production or due to poor hygiene in the processing or distribution chain. Based on the literature review, no such study or finding was documented. However, previous studies reported an association and/or correlation of fecal coliforms with bacterial pathogens in oysters in the USA [38] and in different types of seafood in Thailand [34]. 

Bacterial pathogens may be present at low levels in harvested seafood; however, the level of these pathogens may increase due to poor handling, improper processing, or unsanitary practices. In addition, cross-contamination is the consequence of unhygienic management practices. Cross-contamination can be prevented and controlled through proper handling, processing, transportation and storage. Although the data collected in this study are from a few years ago, these data were compared with data from previous [6,14,22,31] and more recent studies [2,17,34,39]. Furthermore, seafood aquaculture, harvesting and processing guidelines have not been changed significantly over the last several years [37]. 

The results of this study improve our understanding of the microbial contamination of seafood, providing quantitative data for further risk assessment upon which seafood safety policy decisions can be made. These findings are of value to national and international regulatory agencies, the global seafood industry and academia. The results of this study warrant further studies to continuously monitor the microbiological quality and safety of domestic and imported seafood.

## Figures and Tables

**Figure 1 pathogens-12-00187-f001:**
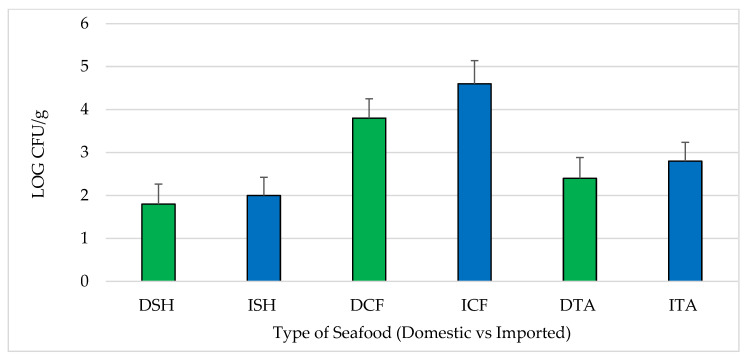
Aerobic plate counts (APC) assembled by type and source of seafood (mean ± standard error). DSH (n = 71) = domestic shrimp, ISH (n = 85) = imported shrimp, DCF (n = 82) = domestic catfish, ICF (n = 60) = imported catfish, DTA (n = 58) = domestic tilapia and ITA (n = 84) = imported tilapia.

**Figure 2 pathogens-12-00187-f002:**
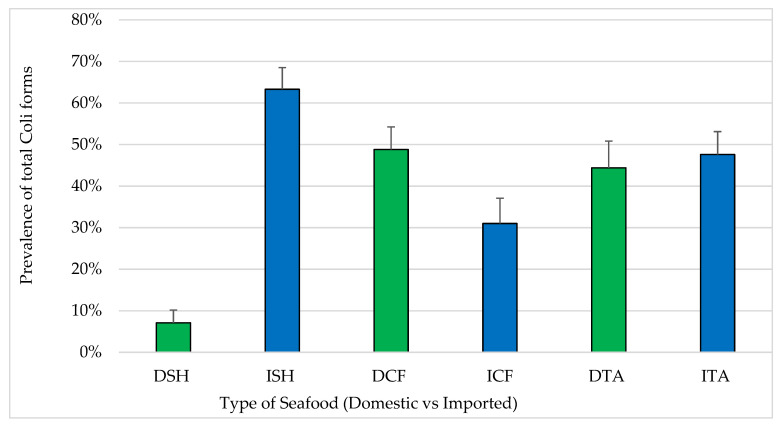
Prevalence of coliforms in seafood (mean ± standard error). Total number of samples: DSH (n = 71) = domestic shrimp, ISH (n = 85) = imported shrimp, DCF (n = 82) = domestic catfish, ICF (n = 60) = imported catfish, DTA (n = 58) = domestic tilapia and ITA (n = 84) = imported tilapia.

**Figure 3 pathogens-12-00187-f003:**
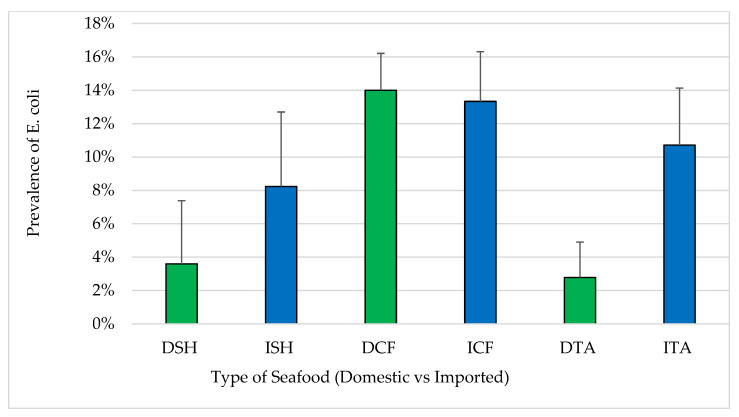
Prevalence of *E. coli* in seafood (mean ± standard error). Total number of samples: DSH (n = 71) = domestic shrimp, ISH (n = 85) = imported shrimp, DCF (n = 82) = domestic catfish, ICF (n = 60) = imported catfish, DTA (n = 58) = domestic tilapia and ITA (n = 84) = imported tilapia.

**Figure 4 pathogens-12-00187-f004:**
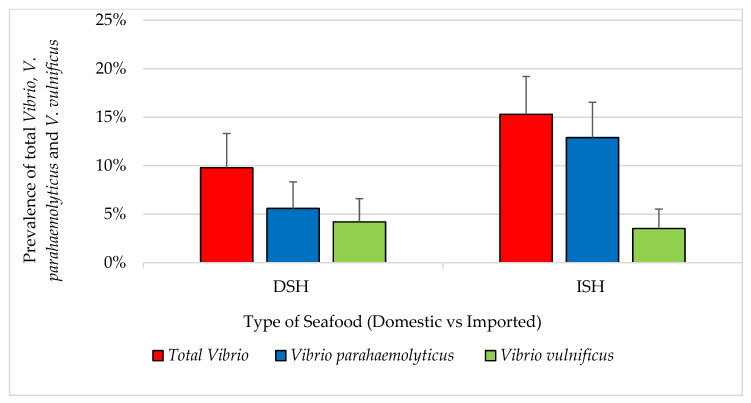
Prevalence of total *Vibrio*, *V. parahaemolyticus* and *V. vulnificus* in seafood (mean ± standard error). DSH (n = 71) = domestic shrimp and ISH (n = 85) = imported shrimp.

**Figure 5 pathogens-12-00187-f005:**
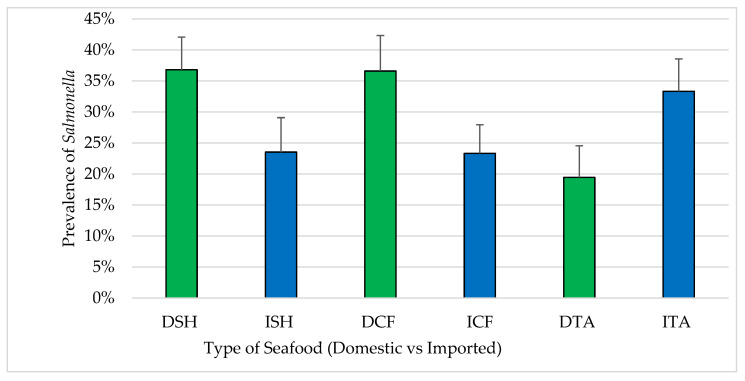
Prevalence of *S.* Typhimurium in seafood (mean ± standard error). DSH (n = 71) = domestic shrimp, ISH (n = 85) = imported shrimp, DCF (n = 82) = domestic catfish, ICF (n = 60) = imported catfish, DTA (n = 58) = domestic tilapia and ITA (n = 84) = imported tilapia.

**Figure 6 pathogens-12-00187-f006:**
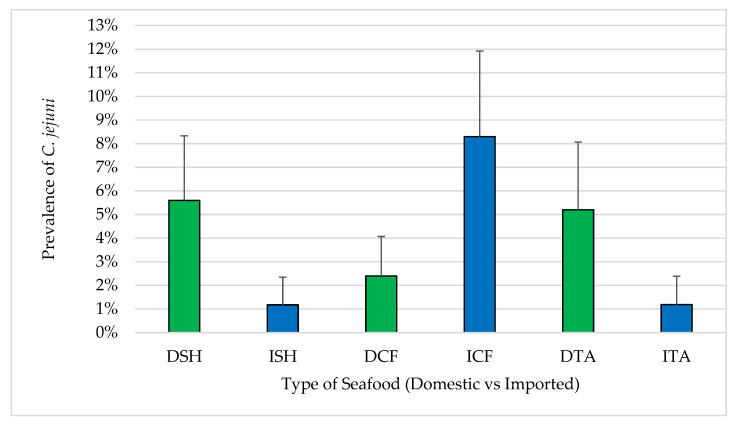
Prevalence of *C. jejuni* in seafood (mean ± standard error). DSH (n = 71) = domestic shrimp, ISH (n = 85) = imported shrimp, DCF (n = 82) = domestic catfish, ICF (n = 60) = imported catfish, DTA (n = 58) = domestic tilapia and ITA (n = 84) = imported tilapia.

**Table 1 pathogens-12-00187-t001:** Microbiological quality of seafood.

Seafood Type	Country of Origin	No. (%) of Samples	No. (%) of Samples That Fell into the Microbiological Quality Category of		
			Good (CFU/g = < 5 × 10^5^)	Marginally Acceptable (CFU/g = < 5 × 10^5^ & > 1 × 10^7)^	Unacceptable (CFU/g = > 1 × 10^7^)
Shrimp	USA	71 (16)	65 (91.5)	2 (2.8)	4 (5.6)
	Ecuador	3 (0.7)	3 (100)	0 (0)	0 (0)
	India	20 (4.5)	19 (95)	1 (5)	0 (0)
	Indonesia	33 (7.5)	30 (90.9)	3 (9.1)	0 (0)
	Thailand	29 (6.6)	27 (93.1)	2 (7.9)	0 (0)
Catfish	USA (*Ictalurus punctatus*)	82 (18.6)	83 (86.4)	9 (9.4)	4 (4.2)
	Vietnam (*Pangasius swai*)	60 (13.6)	53 (88.3)	7 (11.7)	0 (0)
Tilapia	USA	58 (13.1)	54 (93)	4 (5.6)	0 (0)
	China	84 (19.1)	77 (91.7)	7 (8.3)	0 (0)

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
