# Peer review of "Prevalence and Abundance of Bacterial Pathogens of Concern in Shrimp, Catfish and Tilapia Obtained at Retail Stores in Maryland, USA"

_pathogens, 2023, doi:10.3390/pathogens12020187_

Round 1

Reviewer 1 Report

The main comment of this study is that information retrieval is considered relatively not a novelty, since the samples were collected almost 10 years ago. This would not reflect the current situation of seafood contamination and it is difficult to compare with recent studies. In addition, the source of contamination and control and prevention strategies of bacterial contamination should be more thoroughly addressed in the discussion.

More minor comments are indicated below.

-    The title and objectives of the study are not covered by the study. This study examined both qualitative and prevalence of bacteria, so the term quantitative analysis of bacteria or microbiological quantity should be added.

-       The format of subtitles should be carefully checked.

-     Topic 3.3 may not describe the details in this section, so changing the subtitle should be considered.

-       A lower and upper detection limit of the qualitative study should be mentioned.

-       Line 97: What is the purpose of using the data logger. None of the results were mentioned about the data from data logger. The brand and the company should be added.  

-       Please, recheck all abbreviations for bacteria used because there are some errors.

-    The reference format of company for the media used in this study should be checked in lines 106, 114, 120, 138, 152-153, 158, 166 etc.

-       The reference company of the media used in this study should be indicated in the materials and methods.

-       For coliforms and E. coli counts, no bacterial confirmation is performed. The use of Petri film can only confirm these bacteria.

-      In lines 190-193, it is unclear what the further analysis of using storage bacteria at -80C.

-     Inconsistency of materials and methods and result of total Vibrio. The unavailable method was detected, but the result indicated the number of total Vibrio. Therefore, please clarify the materials and methods of these pathogens.

-     The source of bacterial contamination from farm to fork should be addressed.

-  Control and prevention methods should be discussed more to enhance seafood safety. 

-  

Reviewer 2 Report

The authors have conducted retail sampling of shrimp and fish from markets on the Delmarva Peninsula over the period of a year.  They provide prevalence estimates and quantitative levels of indicator organism and pathogens for domestic and imported products. The information should be useful for risk analyses of these products, and for refining safe food handling recommendations.  There are a number of issues that need to be addressed.

Table 1 presents a list of recent outbreaks associated with seafood products.  In its current form it is not useful.  It does not summarize data in a meaningful way.  Although it does provide references, the descriptions of seafood categories are inconsistent, incomplete, and difficult to interpret.  For example, “recalled of many seafood products” is listed for an outbreak from 2021.  Some of the references refer to discrete outbreaks, while others summarize multiple outbreaks over multi-year periods. The impact of outbreaks can be described in text, or a table generated that is in a more consistent and interpretable format.

The order of commodities presented in Figures 1-3 should be consistent.  Figure 3 presents data in a different order. This makes it more difficult to compare across figures. 

The description of preparation of samples is very confusing (section 2.2).  A figure that displayed the aliquots and distribution of tests would be helpful.  Beginning with the three 100g replicate samples, everything gets confusing. Were all tests performed in replicates? What happened to the 200g that wasn’t parceled out into the three 100g replicate samples? Why were 50g samples of shrimp tested, compared to 25g samples of fish?

Lines 47-51 There is no context for the time or location of outbreaks in these lines to compare to the CDC reference in line 45.  It is not clear how much of a background is needed for this introduction.

Lines 69-72 There is no real context presented for looking at Salmonella or Campylobacter in seafood.

Line 85 The samples were collected in 2012-2013.  Is there a reason why the paper is only now being submitted for publication?

Line 88 Were shrimp oversampled during the “second” three-month period?  There could be some additional explanation for the distribution of samples. Since this study was conducted over a 12-month period, was any seasonal analysis of results conducted?

Lines 95-96 It appears that all samples were obtained in a frozen state. Please confirm that this is so.  Did each sample weigh 500 grams at the time of sampling?

Line 97 Did the data logger track sample temperatures?  Were there any unusual results?

Line 102 Were samples homogenized in a blender?  Please elaborate.  

The timing of the study in 2012-2013 resulted in S. Typhimurium isolates being compared by PFGE (data not shown).  However, whole genome sequencing (WGS) has replaced PFGE as the subtyping method of choice.  A couple of the benefits of WGS are serotyping and predicting antimicrobial resistance.  The finding that all Salmonella were S. Typhimurium var 5 is noted as being unusual. However, if isolates were sequenced, it could be useful to clarify the relationship between isolates.  It would be helpful to rule out specimen contamination as a reason for the findings as presented.  WGS could also add a critical dimension of assessing antimicrobial resistance among the various isolates reported.  This is of great importance.  If isolates from this study are still available, sequencing the specimens would be highly desirable.
